# Segment-Aligned Policy Optimization for Multi-Modal Reasoning

**Lei Gao** [1 2 *]   **Zhuoming Li** [3 2 *]   **Mengxi Jia** [2 * †]   **Jiakang Yuan** [1]   **Hongbo Sun** [2]   **Hao Sun** [2 †]   **Xuelong Li** [4]

## Abstract

Existing reinforcement learning approaches for Large Language Models typically perform policy optimization at the granularity of individual tokens or entire response sequences. However, such formulations often misalign with the natural step-wise structure of reasoning processes, leading to suboptimal credit assignment and unstable training in multi-modal reasoning tasks. To bridge this gap, we propose Segment-Aligned Policy Optimization (SAPO), a novel reinforcement learning paradigm that treats coherent reasoning steps, rather than tokens or full sequences as fundamental units of policy update. SAPO introduces a step-wise Markov decision process abstraction over reasoning segments, accompanied by segment-level value estimation, advantage computation, and importance sampling mechanisms that are semantically aligned with reasoning boundaries. Experiments on representative reasoning benchmarks demonstrate that SAPO consistently outperforms token-level and sequence-level policy optimization methods, achieving significant accuracy improvements while exhibiting better training stability and value estimation consistency. Our work underscores the importance of aligning reinforcement learning updates with the intrinsic structure of reasoning, paving the way for more efficient and semantically grounded policy optimization in complex reasoning tasks. Code is available at https://github.com/Graysonicc/SAPO.

## 1. Introduction

Large Language Models (LLMs) have recently exhibited impressive gains in complex reasoning when prompted to produce Chain-of-Thought (CoT) traces, which allows LLMs to address complex tasks by generating a series of intermediate reasoning steps. Despite this progress, the effective optimization of long-form reasoning behaviors remains challenging. Empirically, a model's incorrect final predictions frequently originate from logical deviations in one or a few individual reasoning steps. However, contemporary reinforcement learning (RL) training paradigms lack explicit modeling of reasoning step boundaries and their relative importance. Credit assignment is instead performed at the token or sequence level, which fails to reflect the step-wise structure of reasoning.

For example, as shown in Fig. 1(a), classical RL methods (e.g., PPO (Schulman et al., 2017)) compute token-level advantages using generalized advantage estimation (GAE) for policy updates. In long-CoT reasoning scenarios, sparse terminal rewards must be propagated across a large number of tokens, resulting in high-variance advantage estimates. Furthermore, multiple tokens belonging to the same reasoning step are treated independently and assigned inconsistent learning signals, resulting in gradient updates that fail to reflect the true contribution at the step level. Some tokens within an incorrect reasoning step may be assigned high advantage values, and vice versa. In addition, methods like PPO rely heavily on accurate value estimation for intermediate states. However, the value model is often required to make predictions on intermediate tokens where reasoning is still incomplete and semantic information remains insufficient. This leads to systematic bias in value estimation and further amplifies training noise.

Conversely, group-based sampling methods, such as GRPO (Guo et al., 2025a), leverage relative advantages computed after reward evaluation by comparing samples within each group, eliminating the need for an explicit value model. These methods assign credit uniformly at the sequence level, which has been shown to reduce gradient variance. However, such uniform sequence-level credit assignment prevents the model from distinguishing the contributions of intermediate reasoning steps. For example, when a model reasons correctly in most intermediate steps but produces an incorrect final answer due to a computation or formatting error in the last step, the entire reasoning trajectory is still uniformly penalized, causing many otherwise correct and valuable reasoning steps to be suppressed.

---

[*]Equal contribution † Corresponding author.   [1]Fudan University [2]China Telecom Artificial Intelligence Technology (Beijing) Co., Ltd. [3]Southeast University [4]Institute of Artificial Intelligence, China Telecom.   Correspondence to: Mengxi Jia <mxjia@pku.edu.cn>, Hao Sun <sun.010@163.com>.

*Proceedings of the 43rd International Conference on Machine Learning*, Seoul, South Korea. PMLR 306, 2026. Copyright 2026 by the author(s).

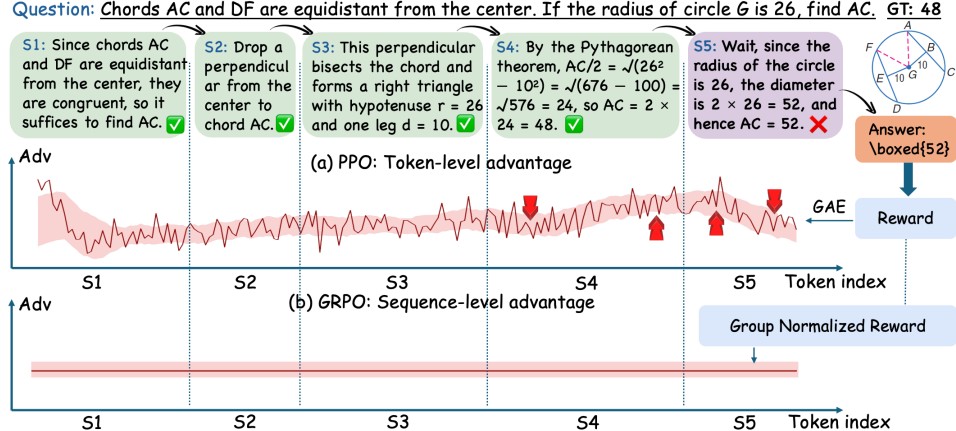

*Figure 1.* Comparison of credit assignment granularity in reinforcement learning for long-CoT. (a) Token-level methods such as PPO assign advantages independently to individual tokens, leading to noisy and inconsistent learning signals within the same reasoning step. (b) Sequence-level methods such as GRPO assign uniform credit across the entire trajectory, failing to distinguish the contributions of individual reasoning steps.

As a result, existing methods lie at two extremes: overly fine-grained token-level modeling (e.g., PPO) and overly coarse sequence-level modeling (e.g., GRPO), both of which fail to align with the step-wise structure of reasoning.

Several recent works (Guo et al., 2025b; Kazemnejad et al., 2025; Liu et al., 2025) have attempted to mitigate these limitations by introducing segment-aware heuristics or modified credit assignment schemes. However, these approaches remain fundamentally grounded in token- or sequence-level optimization, as the policy update (Guo et al., 2025b; Kazemnejad et al., 2025), value estimation (Liu et al., 2025) are still defined over individual tokens or entire sequences, rather than over coherent reasoning steps. As a result, the underlying mismatch between the RL optimization paradigm and the step-wise structure of reasoning persists.

To address this limitation, we propose **SAPO**, a novel policy optimization framework that treats reasoning steps as the primary units of optimization in the Markov Decision Process (MDP (Ouyang et al., 2022)). Specifically, we reformulate the optimization objective under a step-wise MDP perspective, where each action corresponds to generating a coherent reasoning segment. Under this formulation, we define a value function at the level of reasoning steps to capture intermediate progress, addressing the mismatch between token-level value estimation and step-wise reasoning. We then compute advantages over reasoning steps, aligning learning signals with complete reasoning steps instead of individual tokens. Moreover, an importance sampling scheme consistent with this step-wise formulation is introduced to normalize update scales across steps, preventing overly aggressive updates on long reasoning trajectories and thereby maintaining stable policy updates.

To complement the proposed SAPO, we design an adaptive segmentation mechanism that can identify logical boundaries based on high-entropy tokens during generation. Intuitively, high-entropy tokens reflect the moments of uncertainty, at which the model faces multiple competing continuations. By dynamically partitioning the trajectory around these decision points, the adaptive segmentation mechanism can capture step boundaries even when the model does not follow predefined formatting, providing a robust interface between token generation and step-level learning.

Building on the core idea of a step-wise MDP and segment-aligned reinforcement learning, we make the following contributions: (i) We propose **Segment-Aligned Policy Optimization (SAPO)**, a novel reinforcement learning paradigm that introduces a step-wise Markov decision process abstraction over reasoning segments, accompanied by segment-level value estimation, advantage computation, and importance sampling mechanisms. (ii) We introduce an **entropy-based adaptive segmentation mechanism** that dynamically identifies reasoning-step boundaries via high-entropy tokens, and provide empirical evidence supporting its effectiveness compared to existing segmentation strategies. (iii) Experiments on representative reasoning benchmarks demonstrate that SAPO achieves more stable optimization and higher reasoning accuracy than token-level PPO, sequence-level GRPO, and prior segment-level methods.

## 2. Related Work

**Reinforcement Learning with Verifiable Rewards in LLMs.** Reinforcement learning is widely used in LLM post-training to align with human preferences and enhance their reasoning capabilities. Proximal Policy Optimization (PPO, (Schulman et al., 2017)) is widely applied in RL finetuning of LLMs. Direct Preference Optimization (DPO,(Rafailov et al., 2023)) reformulates preference alignment as a binary cross-entropy objective, removing the need

for explicit reward modeling, critics, and on-policy sampling during finetuning. RLOO (Ahmadian et al., 2024) and GRPO (Guo et al., 2025a) eliminate PPO's value network, using the average reward across multiple sampled responses as a policy-gradient baseline; GRPO further estimates advantages by comparing each sample to its group mean reward. Advanced refinements such as GSPO (Zheng et al., 2025), which replaces token-level importance weighting with sequence-likelihood ratios and performs sequence-level clipping in GRPO to reduce variance and stabilize optimization for large MoE models or long responses. Single-stream Policy Optimization (SPO (Xu & Ding, 2025)) advocates a return to the single-stream framework, replacing on-the-fly group baselines with a persistent, KL-adaptive Bayesian value tracker and performing global advantage normalization across the training batch.

**Value Estimation Refinement in Reinforcement Learning.** Value estimation is fundamental to reinforcement learning (RL) for large language models (LLMs), as it provides the basis for credit assignment through the advantage function. In the context of Long Chain-of-Thought (Long-CoT), value estimation faces unique challenges such as initialization bias and reward signal decay. VC-PPO (Yuan et al., 2025) introduces Value-Pretraining to align the critic with the initial policy and Decoupled-GAE to ensure reward signals propagate across long sequences without loss, resolving the collapse issue of PPO in long-sequence tasks. VAPO (Yue et al., 2025) proposes Length-adaptive GAE, which dynamically adjusts the GAE decay parameter based on response length to prevent bootstrapping errors from dominating long-sequence estimates. Some research has addressed these inaccuracies by incorporating higher-fidelity value signals. By leveraging the deterministic nature of language environments to reset and sample auxiliary rollouts from intermediate states, VinePPO (Kazemnejad et al., 2025) achieves precise credit assignment with unbiased Monte Carlo (MC) estimation instead of relying on a separate critic model. Similarly, Direct Value Optimization (Zhang et al., 2025) shifts from relative preference labels to absolute stepwise value signals by employing a mean squared error (MSE) loss to align the policy model directly with target values derived from Monte Carlo Tree Search (MCTS) or outcome value models. Together, these methods illustrate a transition from learned, potentially biased critics toward more reliable, calibrated, or sampling-based value estimation frameworks for complex reasoning.

## 3. Preliminaries

To facilitate a clear and precise formulation of our segment-level optimization algorithm, in this section, we first introduce the fundamental concepts, notation, and optimization objectives of reinforcement learning (RL). Then, Proximal Policy Optimization (PPO) and its key components will be introduced.

**Token-Level MDP for Language Generation.** Autoregressive language generation is typically modeled as a Markov Decision Process (MDP), defined by $\mathcal{M} = (\mathcal{S}, \mathcal{A}, P, r, \gamma)$, where the decision process unfolds at the token level. Specifically, given an input prompt $x$, a response is generated sequentially as a sequence of tokens $y = (y_1, \ldots, y_T)$. At time step $t$, the state $s_t \in \mathcal{S}$ consists of the prompt together with all previously generated tokens. The action $a_t \in \mathcal{A}$ corresponds to selecting the next token from a fixed vocabulary according to a policy $\pi_\theta(a_t \mid s_t)$, where $\theta$ is the model parameters. $P$ is a state transition function, denoting the probability of transitioning from state $s_t$ to state $s_{t+1}$ after taking action $a_t$. This transition is deterministic because the environment is known and the next state is uniquely determined by appending the selected token to the current context, i.e., $P(s_{t+1} \mid s_t, a_t) = 1$. The reward function $r(s_t, a_t)$ provides a scalar feedback signal for the generation process.

**Learning Objective in Policy Optimization.** The objective of reinforcement learning is to learn a policy $\pi_\theta$ that maximizes the expected cumulative reward over trajectories sampled from the policy:

$$J_{PPO}(\theta) = \mathbb{E}_{\tau \sim \pi_\theta} \left[ \sum_{t=0}^{T} \gamma^t r_t \right]. \tag{1}$$

where $T$ is the total number of decision steps, and $r_t$ is the token-level reward calculated by the reward function. In RLVR (Huang et al., 2026), rewards are verifiable and sparse, with $r_t = 0$ for $t < T$ and a terminal reward determined by the verifier.

**Proximal Policy Optimization.** PPO updates the policy by optimizing a clipped surrogate objective that constrains the magnitude of policy updates between successive iterations. Let $\pi_\theta$ denote the current policy and $\pi_{old}$ the policy used to collect trajectories. The importance sampling ratio is defined as:

$$r_t(\theta) = \frac{\pi_\theta(a_t \mid s_t)}{\pi_{\theta_{old}}(a_t \mid s_t)}. \tag{2}$$

The PPO objective is given by:

$$\mathcal{L}(\theta) = \mathbb{E}_t \left[ \min\left( r_t(\theta) A_t, \ \text{clip}\left( r_t(\theta), 1 - \epsilon, 1 + \epsilon \right) A_t \right) \right]. \tag{3}$$

where $A_t$ denotes an advantage estimate and $\epsilon > 0$ is a clipping threshold that limits the change in action probabilities. $A_t$ is computed using Generalized Advantage Estimation (GAE):

$$A_t = \sum_{l=0}^{T-t-1} (\gamma \lambda)^l \delta_{t+l}. \tag{4}$$

where $\delta_t = r_t + \gamma V(s_{t+1}) - V(s_t)$ is the temporal-difference error at time step $t$, and $V(s_t)$ represents the state value, estimated by the value function $V(s)$.

# 4. Method

To address the mismatch between existing policy optimization formulations and the step-wise structure of reasoning, we propose Segment-Aligned Policy Optimization (SAPO), treating reasoning steps, rather than individual tokens or entire sequences, as the fundamental units of policy optimization, which allows reasoning to be modeled as a step-wise Markov decision process. We illustrate SAPO in Fig. 2 and present a pseudo-code implementation in Algorithm. 1.

## 4.1. Segment-Aligned Policy Optimization

To align policy optimization with the step-wise structure of reasoning, the learning signal should ideally be assigned at the granularity of reasoning steps, rather than being fragmented at the token level or uniformly applied at the sequence level. Accordingly, we introduce a *segment-level temporal abstraction* by partitioning the response $y$ into $M$ coherent segments:

$$y = \big[z_1 \,\|\, z_2 \,\|\, \cdots \,\|\, z_M\big], \; z_m = \big(y_{b_m}, \ldots, y_{e_m}\big). \quad (5)$$

where $1 = b_1 \leq e_1 < b_2 \leq \cdots < e_M = T$, and $\|$ denotes concatenation. We treat each segment $z_m$ as a high-level semantic unit for policy optimization, and define the segment-level "state" as the segment boundary prefix $s_m := (x, y_{\leq e_{m-1}})$ with $e_0 := 0$.

### 4.1.1. STEP-WISE MDP FORMULATION

We model the reasoning process as a step-wise Markov decision process (MDP), where each state corresponds to a coherent reasoning segment and transitions occur only at segment boundaries. Formally, a reasoning episode starts from an initial state

$$s_1 = \{x\},$$

where $x$ denotes the input prompt. At step $m$, the state is defined as

$$s_m = \{x, z_{<m}\},$$

where $z_{<m} = [z_1, \ldots, z_{m-1}]$ is the sequence of previously generated reasoning segments. The policy $\pi_\theta$ selects an action $a_m$, corresponding to generating the next reasoning step $z_m$, according to $\pi_\theta(a_m \mid s_m)$.

State transitions are deterministic: the next state $s_{m+1}$ is fully determined by appending the generated segment $y_m$ to the history. The episode terminates when a special end condition is reached (e.g., generation of the final answer).

Following the RLVR setting, rewards are provided only at the end of the episode based on the correctness of the final answer, and no intermediate rewards are assigned to individual segments. Credit assignment to intermediate reasoning steps is therefore handled implicitly through segment-level value estimation and advantage computation, rather than explicit step-wise rewards. This step-wise MDP abstraction serves as a structured interface between token-level generation and step-level policy optimization, enabling credit assignment to align with the semantic structure of reasoning.

### 4.1.2. STEP-WISE VALUE ESTIMATION AND ADVANTAGE COMPUTATION

Under this formulation, we learn a value function $V(s_m)$ that predicts the expected outcome reward from the segment state $s_m$. Let $r_m$ be the segment-level reward at boundary $e_m$. For outcome-only tasks, $r_m = 0$ for $m < M$ and $r_M = r(x, y)$. We define temporal-difference (TD) errors over segment states as:

$$\delta_m = r_m + \gamma V(s_{m+1}) - V(s_m), \;\; m = 1, \ldots, M. \quad (6)$$

Advantage estimates and return targets are obtained by accumulating TD errors across segments:

$$\hat{A}_m = \sum_{\ell=0}^{M-m} (\gamma\lambda)^\ell \delta_{m+\ell}, \;\; \hat{R}_m = \hat{A}_m + V(s_m). \quad (7)$$

We assign the step-wise advantage $\hat{A}_m$ uniformly to all tokens within segment $z_m$, i.e., $\hat{A}_t := \hat{A}_m$ for all $t \in [b_m, e_m]$, ensuring that tokens belonging to the same reasoning step receive consistent learning signals.

### 4.1.3. IMPORTANCE SAMPLING OVER STEPS

To ensure stable and well-calibrated policy updates under the step-wise formulation, we introduce an off-policy correction defined over reasoning steps. Specifically, for samples generated by an old policy $\pi_{\theta_{\text{old}}}$, we define the importance ratio for segment $z_m$ as a geometric mean of token-level likelihood ratios:

$$
\begin{aligned}
s_m(\theta) :&= \left( \frac{\pi_\theta(z_m \mid s_m)}{\pi_{\theta_{\text{old}}}(z_m \mid s_m)} \right)^{\frac{1}{|z_m|}} \\
&= \exp\left( \frac{1}{|z_m|} \sum_{t=b_m}^{e_m} \log \frac{\pi_\theta(y_t | x, y_{<t})}{\pi_{\theta_{\text{old}}}(y_t | x, y_{<t})} \right).
\end{aligned}
\quad (8)
$$

Although the geometric-mean formulation is not a strictly unbiased importance-sampling estimator for the raw segment-level action probability, we view it as a controlled bias–variance tradeoff for variable-length reasoning. The raw segment ratio multiplies token-level likelihood ratios

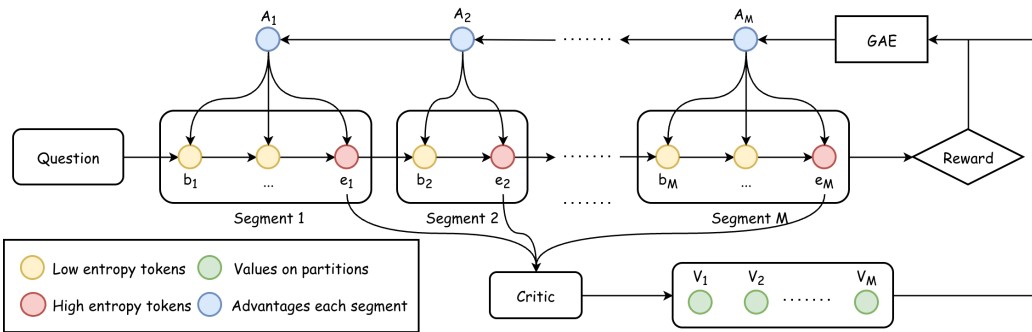

*Figure 2.* Overview of Segment-Aligned Policy Optimization (SAPO). The response is partitioned into coherent reasoning steps using entropy-aware boundaries. SAPO performs policy optimization and value learning over these reasoning steps, ensuring that tokens within the same step are updated with consistent learning signals.

across the entire segment, which can lead to rapidly increasing variance as segment length grows. In contrast, the geometric mean measures the average per-token policy shift within a reasoning step, yielding a length-normalized correction that is more stable across segments of different sizes. This choice therefore deliberately introduces a bounded approximation bias in exchange for substantially improved numerical stability and lower variance in policy updates. A formal proof of the resulting bias bound is provided in the Appendix D.

### 4.1.4. LEARNING OBJECTIVES.

SAPO optimizes a clipped surrogate objective defined over reasoning steps. The policy objective can be defined as:

$$J(\theta) = \mathbb{E}_{x \sim \mathcal{D}, \, y \sim \pi_{\theta_{\text{old}}}(\cdot|x)} \left[ \frac{1}{T} \sum_{t=1}^{T} \min \left( s_{m(t)}(\theta) \, \hat{A}_{m(t)}, \right. \right.$$

$$\left. \left. \text{clip}(s_{m(t)}(\theta), 1 - \epsilon, 1 + \epsilon) \, \hat{A}_{m(t)} \right) \right].$$

(9)

where $m(t)$ denotes the index of the segment that token $t$ belongs to.

In parallel, we learn a value function $V(s_m)$ and update it *only at segment boundaries* to match the segment-level temporal abstraction. Given the returns $\hat{R}_m$ (Eq. (7)), the value objective is defined as:

$$\mathcal{L}_V(\phi) = \mathbb{E}_{x \sim \mathcal{D}, \, y \sim \pi_{\theta_{\text{old}}}(\cdot|x)} \left[ \frac{1}{M} \sum_{m=1}^{M} \left( V_\phi(s_m) - \hat{R}_m \right)^2 \right].$$

(10)

### 4.2. Entropy-based Adaptive Segmentation Mechanism

To operationalize reasoning steps without external annotations, inspired by prior work (Wang et al., 2025), we use token entropy

$$H_t = -\sum_v \pi_\theta(v \mid x, y_{<t}) \log \pi_\theta(v \mid x, y_{<t}) \qquad (11)$$

as an uncertainty signal. Given the entropy sequence $\{H_t\}_{t=1}^{T}$, we identify segment boundaries by selecting token indices whose entropy ranks in the top-$k\%$ among all tokens in the sequence, and always include $T$ as the final boundary to terminate the last segment.

To justify entropy-based segmentation, we analyze the relationship between token entropy and value discontinuities $\Delta V_t = |V_t - V_{t+1}|$. In standard reinforcement learning, the state-value $V^\pi(s)$ denotes the expected discounted return under policy $\pi$ starting from state $s$ (Sutton et al., 2018). PPO and other actor–critic methods learn a parametric critic function $V(s)$ to approximate this expectation. High-entropy tokens correspond to states where multiple continuations compete, reflecting elevated uncertainty over future outcomes. When these alternatives imply substantially different future returns, transitioning across such points can induce a larger change in the critic's value estimate, resulting in higher expected value discontinuities.

We formalize $\mathcal{A}$ as the set of high-entropy tokens whose entropy lies in the top-$q$ quantile, and define $\mathcal{D}$ as the set of branching tokens with large value discontinuities. Their association is quantified by the enrichment ratio

$$Lift(\mathcal{A}, \mathcal{D}) = \frac{\mathbb{P}(\mathcal{D} \mid \mathcal{A})}{\mathbb{P}(\mathcal{D})}, \qquad (12)$$

with uncertainty estimated via sample-level bootstrap to obtain 95% confidence intervals.

As shown in Fig. 3, $Lift$ increases monotonically with the entropy quantile threshold $q$ and exceeds 1 in the high-entropy tail, indicating that decision points are significantly enriched in regions of elevated uncertainty. Moreover, stage-averaged $Lift$–$q$ curves exhibit a clear upward shift from early to late training (we froze the policy model to guarantee

accurate value estimation), with the regime where $Lift > 1$ progressively extending toward lower entropy thresholds.

Taken together, these results demonstrate that entropy-based segmentation yields adaptive reasoning segments that consistently align with decision points and semantic transitions, providing a principled interface between token-level generation and step-wise optimization. In this condition, SAPO can be viewed as an adaptive temporal abstraction in the spirit of Semi-MDPs (Sutton et al., 1999), where variable-length reasoning chunks replace fixed token-level transitions, thus reducing intra-step variance while preserving sensitivity to key decisions, achieving a better bias–variance tradeoff. A more comprehensive analysis of our entropy-based segmentation strategy, including its connection to value discontinuities and the target of variance reduction, is provided in Appendix C.

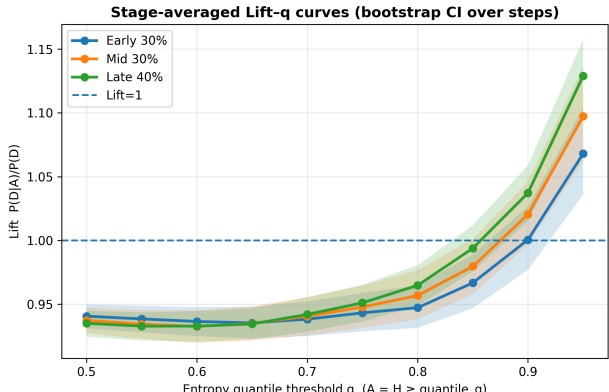

*Figure 3.* **Stage-averaged $Lift$–$q$ curves across training.** $Lift$ increases with the entropy threshold and shifts upward from early to late training stages, with the $Lift > 1$ region expanding toward lower $q$. Shaded areas denote 95% bootstrap confidence intervals over steps. Moreover, the smoothed mean $Lift$ exhibits a positive monotonic trend over training (Spearman $\rho = 0.15$; bootstrap 95% CI for the linear slope excludes zero). This indicates that decision points become increasingly concentrated in high-entropy regions as training proceeds.

### 4.3. Gradient Analysis

We derive the gradient of the SAPO objective as follows (clipping is omitted for brevity):

$$\nabla_\theta J(\theta) = \mathbb{E}_{x \sim \mathcal{D}, \, y \sim \pi_{\theta_{old}}(\cdot|x)} \left[ \frac{1}{T} \sum_{t=1}^{T} s_{m(t)}(\theta) \, \hat{A}_{m(t)} \right.$$
$$\left. \cdot \frac{1}{|z_{m(t)}|} \sum_{i=b_{m(t)}}^{e_{m(t)}} \nabla_\theta \log \pi_\theta(y_i \mid x, y_{<i}) \right].$$

(13)

From Eq. (13), SAPO preserves the token-level policy gradient structure of PPO, while replacing token-level importance ratios with segment-level ratios shared across tokens within the same reasoning step. As a result, all tokens in a segment

are scaled by an identical importance weight and advantage, which enforces consistent credit assignment within each reasoning step. Moreover, the averaged log-likelihood ratio introduces an implicit normalization with respect to segment length, preventing the importance ratio and its gradient from growing with the number of tokens. Compared to token-level PPO, this design reduces within-step gradient variance while remaining more fine-grained than sequence-level methods such as GRPO.

## 5. Experiment

### 5.1. Experimental Setups

We conduct experiments on both multi-modal reasoning benchmarks and text-only reasoning benchmarks. For multi-modal reasoning, we use Qwen2.5-VL-Instruct (Bai et al., 2025) as the base model. Models at both scales (3B and 7B) are trained using only the Geo3K (Lu et al., 2021) training split and evaluated on Geo3K (test), LogicVista (Xiao et al., 2024), MathVerse (Zhang et al., 2024), WeMath (Qiao et al., 2025), MathVista (Lu et al., 2023), and DynaMath (Zou et al., 2024). We compare SAPO against several representative reinforcement learning baselines, including GRPO, PPO, and its variant Value-Calibrated PPO (VC-PPO). For text-only reasoning, we evaluate our method on two representative mathematical reasoning models, RhoMath-1.1B (Lin et al., 2024) and DeepSeekMath-7B (Shao et al., 2024), to assess its effectiveness. The training data consists of the training splits of MATH (Hendrycks et al., 2021) and GSM8K (Cobbe et al., 2021), and evaluation is performed on their corresponding test sets (GSM8K-Test and MATH500). We compare against several representative baselines, including sequence-level methods RLOO and GRPO, token-level method PPO, RL-free methods RestEM (Singh et al., 2023) and DPO+ (Pal et al., 2024), segment-level method VinePPO and SPO. For the entropy-based adaptive segmentation mechanism, we set $k$ to 30. All methods share the same outcome reward functions and optimization hyperparameters to ensure a fair and controlled comparison, with detailed training configurations and hyperparameters provided in the Appendix. F.

### 5.2. Main Results

**Results on Multi-modal Reasoning** Tab. 1 shows the main experimental results on six representative multi-modal reasoning benchmarks. Across both model scales (3B and 7B), SAPO consistently outperforms GRPO, PPO, and VC-PPO, indicating that the proposed segment-aligned optimization is effective across different model capacities rather than being specific to a particular scale. Specifically, SAPO achieves substantially stronger performance on the Geo3K test set for both model sizes (e.g., +24.46% for 3B and +10.15% for 7B compared to PPO), reflecting more effec-

*Table 1.* Performance comparison of Qwen2.5-VL models trained with different reinforcement learning strategies on six multi-modal reasoning benchmarks.

| Benchmark | Geo3K | LogicVista | MathVerse | WeMath | MathVista | DynaMath | Avg. |
|---|---|---|---|---|---|---|---|
| **3B Models** | | | | | | | |
| Qwen2.5-VL-3B | 27.29 | 34.45 | 24.75 | 21.81 | 61.60 | 10.97 | 30.15 |
| +GRPO | 40.60 | 40.72 | 36.67 | 29.43 | 63.00 | 17.37 | 37.97 |
| +PPO | 21.80 | 39.15 | 26.52 | 25.90 | 63.80 | 9.98 | 31.19 |
| +VC-PPO | 34.94 | 37.81 | 32.36 | 32.67 | 62.90 | 10.78 | 35.24 |
| **+SAPO** | 46.26 | 39.60 | 37.56 | 36.19 | 65.40 | 16.37 | **40.23** |
| **7B Models** | | | | | | | |
| Qwen2.5-VL-7B | 42.60 | 39.15 | 30.58 | 32.67 | 69.30 | 13.37 | 37.94 |
| +GRPO | 51.58 | 45.86 | 44.80 | 41.05 | 69.70 | 22.75 | 45.96 |
| +PPO | 42.60 | 42.73 | 39.72 | 41.52 | 69.80 | 17.96 | 42.39 |
| +VC-PPO | 48.75 | 45.41 | 41.37 | 40.48 | 68.40 | 20.76 | 44.20 |
| **+SAPO** | 52.75 | 47.30 | 45.81 | 42.86 | 70.70 | 21.96 | **46.90** |

tive utilization of learning signals enabled by the proposed optimization formulation. Moreover, the performance gains become more pronounced on challenging benchmarks such as WeMath, which require long-horizon, multi-step reasoning: SAPO improves accuracy from 29.43% to 36.19% on the 3B model and from 41.05% to 42.86% on the 7B model compared to GRPO. This pattern suggests that aligning policy optimization with step-wise reasoning is particularly beneficial for harder problems that require long-horizon, multi-step reasoning. Additional results in Tab. 5 in Appendix E demonstrate SAPO's strong generalization to advanced reasoning models.

**Results on Text-only Reasoning** Fig. 4 presents the results on GSM8K and MATH500, evaluating text-only mathematical reasoning under different RL optimization methods. SAPO consistently achieves the best performance across both benchmarks, improving over prior approaches such as GRPO and PPO under both 3B and 7B settings. On GSM8K, SAPO yields steady gains over token-level or sequence-level baselines (e.g., +5.3% over PPO and +9.4% over GRPO on DeepSeekMath-7B), and further outperforms prior segment-level methods such as VinePPO (+4.1%) and SPO (+6.0%). More notably, on the more challenging MATH500 benchmark, SAPO still shows marked improvements over both token-level and sequence-level methods (e.g., +7.5% over PPO and +7.8% over GRPO on RhoMath-1.1B), while consistently surpassing VinePPO (+2.6%) and SPO (+4.8%). The consistent improvements over VinePPO and SPO suggest that SAPO's gains stem from its overall optimization formulation, leading to more stable and effective training dynamics.

## 5.3. Ablation Studies

In this subsection, we present ablation studies that examine the contributions of the key design components in SAPO, including importance sampling over reasoning segments,

different segmentation strategies, and the hyperparameter $k$ used in entropy-based segmentation.

*Table 2.* Ablation study on the effect of step-wise IS.

| Method | Geo3K | LogicVista | MathVerse |
|---|---|---|---|
| SAPO (naive-IS) | 47.92 | 44.07 | 43.91 |
| SAPO (sw-IS) | **52.75** | **47.30** | **45.81** |

**Ablation on Importance Sampling over Reasoning Steps.** Tab. 2 reports the ablation results for importance sampling over reasoning steps. Replacing token-level importance sampling with segment-based importance sampling yields consistent performance gains across all evaluated benchmarks, indicating that importance sampling is a critical component of SAPO. In particular, SAPO improves accuracy by +4.83% on Geo3K, +3.23% on LogicVista, and +1.90% on MathVerse compared to its variant with token-level importance sampling (naive-IS). These gains suggest that performing importance sampling at the granularity of reasoning steps leads to better-calibrated policy updates, reducing biased or overly aggressive updates caused by distribution shifts between the behavior policy and the updated policy. The consistent improvements across benchmarks of varying difficulty further demonstrate that importance sampling aligned with reasoning steps is essential for stable and effective policy optimization in SAPO.

*Table 3.* Ablation study on Segmentation Strategy.

| Strategy | Geo3K | LogicVista | MathVerse |
|---|---|---|---|
| Newline-based | 41.60 | 46.09 | 42.13 |
| Step-based | 41.60 | **48.10** | 42.77 |
| Prob-based | 44.43 | 44.97 | 41.62 |
| Entropy-based | **52.75** | 47.30 | **45.81** |

**Ablation on Segmentation Strategy.** Tab. 3 compares different segmentation strategies within SAPO. We consider three representative approaches: (1) Newline-based

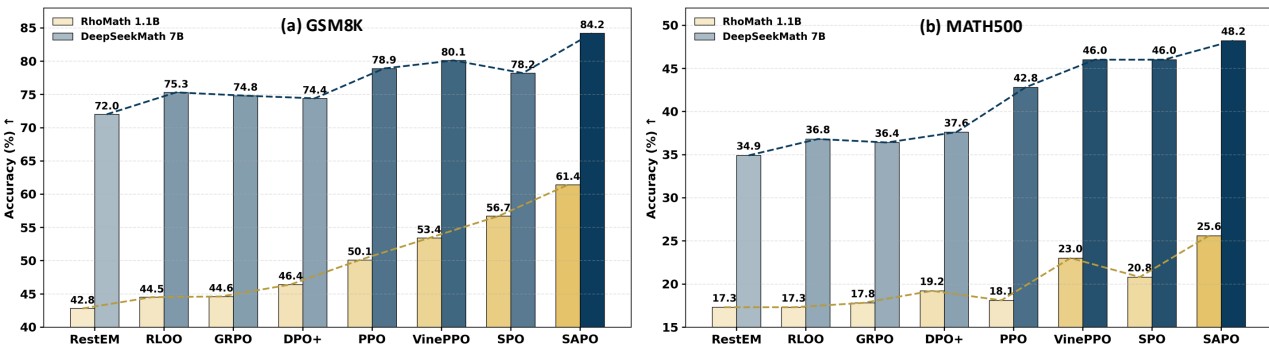

*Figure 4.* Comparison of text-only reasoning performance on GSM8K and MATH500 for RhoMath 1.1B and DeepSeekMath 7B under different reinforcement learning strategies. Baseline results are from (Guo et al., 2025b; Kazemnejad et al., 2025).

segmentation, which splits the generated response according to newline characters (Kazemnejad et al., 2025; Zhang et al., 2025); (2) Step-based segmentation, which explicitly instructs the model to output intermediate reasoning in a predefined step-by-step format (e.g., "Step 1", "Step 2", etc.) and treats each prompted step as a segment (Luo et al., 2023); (3) Probability-based segmentation, which defines segments by accumulating a fixed number of low probability tokens (Guo et al., 2025b). and (4) the proposed entropy-based adaptive segmentation. Among these strategies, entropy-based segmentation achieves the best overall performance. These results demonstrate that adaptively identifying segment boundaries based on model uncertainty leads to more effective utilization of learning signals, highlighting the importance of flexible and model-aware segmentation strategies in policy optimization. We further compare different segmentation strategies across multiple cross-domain benchmarks. Results in Tab. 6 in Appendix E show that our entropy-based adaptive segmentation strategy has better generalization.

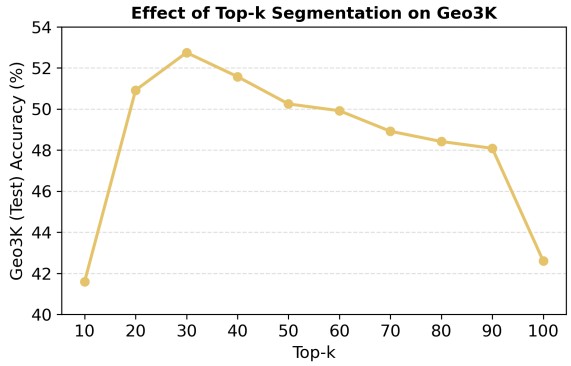

*Figure 5.* Effect of the top-$k\%$ ratio in entropy-based segmentation on Geo3K test accuracy.

**Sensitivity to Hyperparameter k in Entropy-based Segmentation.** Fig. 5 investigates the impact of the top-k

ratio used in the entropy-based segmentation mechanism. We observe that the performance consistently improves as top-k increases from 10 to 30, and then gradually degrades when further increasing top-k beyond this range. This trend aligns with recent findings that only a minority of high-entropy tokens act as critical forks in the reasoning trajectory, while the majority of low-entropy tokens mainly follow already determined paths and contribute limited learning signal (Wang et al., 2025). When top-k is too small (e.g., 10), the segmentation mechanism fails to cover all decision-critical tokens, resulting in insufficient exploration and under-optimized reasoning steps. However, a large top-k introduces an increasing number of low-entropy tokens into the segmentation process, which obscures genuinely decision-critical boundaries and reduces the effectiveness of step-wise policy optimization. This observation further suggests that entropy-based segmentation is most effective when it selectively targets decision-critical tokens rather than uniformly increasing segmentation density, echoing the finding that high-entropy minority tokens dominate learning dynamics in RL-based reasoning models. We transfer the same top-k (30%) setting without retuning to broader benchmarks: scientific reasoning (ScienceQA (Lu et al., 2022)), commonsense multimodal reasoning (MMMU (Yue et al., 2024)), and comprehensive visual-language reasoning (MMBench (Liu et al., 2024), MMStar (Chen et al., 2024)). As shown in Tab. 4, SAPO achieves the best overall performance, indicating that its effectiveness does not rely on dataset-specific tuning of k, but generalizes well across different domains and task types.

*Table 4.* Performance on more general reasoning tasks using Qwen2.5VL-7B as the base model.

| Method | ScienceQA | MMMU | MMBenchV11 | MMStar | Avg. |
|---|---|---|---|---|---|
| Base | 88.34 | 52.67 | 81.23 | 62.73 | 71.24 |
| +GRPO | 88.34 | 51.67 | 81.35 | 63.67 | 71.26 |
| +PPO | 88.79 | 50.44 | 81.09 | 64.00 | 71.08 |
| +SAPO | **90.08** | **53.22** | **81.61** | **64.53** | **72.36** |

### 5.4. Analysis on Value Model Training Progress.

Fig. 6 illustrates the training dynamics of the value model under SAPO and the VC-PPO baseline. Both methods exhibit a rapid decrease in value loss during the early training stage, indicating that the value function quickly learns a coarse estimate of returns. However, as training proceeds, the value loss under SAPO consistently converges to a lower level and remains more stable, whereas VC-PPO shows higher residual loss and noticeable fluctuations throughout training. This suggests that SAPO provides a more coherent and less noisy learning signal for value estimation. We attribute this behavior to SAPO's segment-aligned optimization, which restricts value updates to semantically meaningful reasoning steps rather than individual tokens, thereby reducing variance introduced by token-level noise. In contrast, VC-PPO relies on token-level value supervision, causing the value model to fit many intermediate tokens that lack clear semantic meaning, which leads to less stable convergence. Overall, these results indicate that aligning value learning with reasoning steps improves both the stability and accuracy of value estimation.

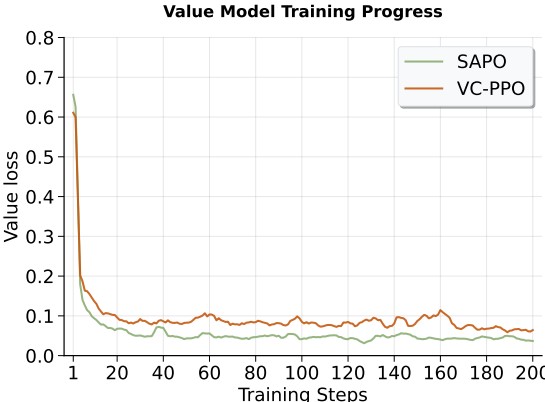

*Figure 6.* Comparison of value model training progress between SAPO and VC-PPO on Geo3K.

## 6. Conclusion

In this work, we analyzed the limitations of existing reinforcement learning approaches for reasoning in MLLMs and identified a fundamental mismatch between their optimization granularity and the step-wise structure of reasoning. To address this issue, we proposed Segment-Aligned Policy Optimization (SAPO), which aligns value estimation, advantage computation, and policy updates with reasoning steps under a step-wise MDP abstraction. By further introducing an entropy-based adaptive segmentation mechanism, SAPO identifies reasoning step boundaries based on model uncertainty during generation. Extensive experiments demonstrate that SAPO achieves more stable optimization and consistently improves reasoning performance over token-level,

sequence-level baselines, and prior segment-level methods. We hope that this work can contribute to policy optimization in reinforcement learning and inspire future research on structured reasoning.

## Impact Statement

This paper presents methodological contributions aimed at improving policy optimization in reinforcement learning, particularly for structured reasoning tasks. The proposed approach focuses on aligning optimization objectives with the inherent structure of model reasoning, with the goal of enhancing training stability and effectiveness. The techniques introduced in this work are general and algorithmic in nature, and are intended to support the development of more reliable learning systems. We do not anticipate any immediate negative societal impacts arising from this research, nor do we identify specific broader societal consequences that require separate discussion.

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

---

**Algorithm 1** Segment-Aligned Policy Optimization (SAPO)

---

**Input:** dataset $\mathcal{D}$, initial policy $\pi_\theta$, value function $V_\phi$, number of epochs $E$, number of mini-batches $B$, clip parameter $\epsilon$, discount factor $\gamma$, decay parameter $\lambda$, segmentation hyperparameter $k$
**Output:** optimized policy $\pi_\theta$, optimized value function $V_\phi$
**for** $e = 1$ **to** $E$ **do**
    Sample a batch of queries $x \sim \mathcal{D}$
    Generate responses $y \sim \pi_{\theta_{\text{old}}}(\cdot \mid x)$ and record token entropies $\{H_t\}$
    Select segmentation boundaries using a top-$k\%$ entropy criterion and include $T$ as the final boundary
    Partition each response into a variable number of segments $\{z_{i,m}\}_{m=1}^{N_i}$
    Construct segment states $s_{i,m} = (x_i, y_{i,\leq e_{m-1}})$
    Compute rewards $\{r_{i,m}\}$ at segment boundaries
    Compute TD errors $\{\delta_{i,m}\}$, advantages $\{\hat{A}_{i,m}\}$, and returns $\{\hat{R}_{i,m}\}$ over segments
    Split the batch of trajectories (each with its segments) into $B$ mini-batches
    **for** $b = 1$ **to** $B$ **do**
        Sample a mini-batch of trajectories $\{(x_i, y_i, \{s_{i,m}, z_{i,m}, \hat{A}_{i,m}, \hat{R}_{i,m}\}_{m=1}^{N_i})\}$
        Compute probability ratios $r_{i,m}(\theta)$ for all segments in the mini-batch
        Compute the clipped policy objective by aggregating over segments and assigning $\hat{A}_{i,m}$ to tokens within $z_{i,m}$
        Update policy parameters $\theta$ by maximizing the clipped objective
        Compute the value loss at segment boundaries
        Update value parameters $\phi$ by minimizing the value loss
    **end for**
**end for**
**return** $\pi_\theta, V_\phi$

---

# A. Benchmark Settings.

**Multi-modal Benchmarks** **LogicVista** emphasizes logical and compositional reasoning in visually grounded settings. The full test set contains approximately 450 samples. **MathVerse** is designed to assess general mathematical reasoning across multiple modalities and input settings. In this work, we use the MathVerse_MINI Test split under the Vision Only setting, which contains around 700 samples. **WeMath** targets fine-grained mathematical reasoning with strict answer verification. We evaluate on the Test Mini split, which includes around 1,740 samples, and report Score (Strict) as the primary metric. **MathVista** is a large-scale visual mathematical reasoning benchmark that integrates natural images with algebraic, geometric, and logical reasoning. In this work, we use the MathVista_MINI test split, containing around 1,000 samples. **DynaMath** is a large-scale benchmark constructed to evaluate robustness and generalization in mathematical reasoning. It consists of 501 original questions, each expanded into 10 distinct variants, yielding approximately 5,000 test samples in total.

**Text-only Benchmarks** **MATH** consists of challenging problems drawn from high-school mathematics competitions, covering a broad range of topics such as algebra, geometry, number theory, and probability. For our experiments, we adopt the OpenAI split (Lightman et al., 2023), which includes 12,500 training problems and 500 test problems. The final answer is explicitly marked using the \boxed{} notation. **GSM8K** is a collection of high-quality grade-school mathematics problems that require basic arithmetic and elementary algebraic reasoning. Although simpler than the MATH, GSM8K remains a widely adopted benchmark for evaluating the mathematical reasoning capabilities of large language models. The dataset consists of 7,473 training problems and 1,319 test problems. The final answer is typically an integer explicitly marked in the solution (e.g., by the delimiter "####"), enabling reliable automatic correctness checking.

# B. Additional Training Progress Analysis.

**Analysis on Entropy.** Figure 7 left part illustrates the evolution of entropy during training for PPO and SAPO. PPO suffers from a much faster entropy collapse, with entropy quickly approaching near-zero values in early training stages. In contrast, SAPO maintains higher entropy throughout training, suggesting a more stable exploration behavior induced by the policy optimization at the level of reasoning steps.

**Analysis on Response Length.** Figure 7 right part illustrates the evolution of response length during training for PPO and SAPO. PPO exhibits a rapid collapse in response length during early training stages, quickly converging to very short outputs. In contrast, SAPO maintains longer responses throughout training, indicating that policy optimization at the level of reasoning steps encourages sustained multi-step generation.

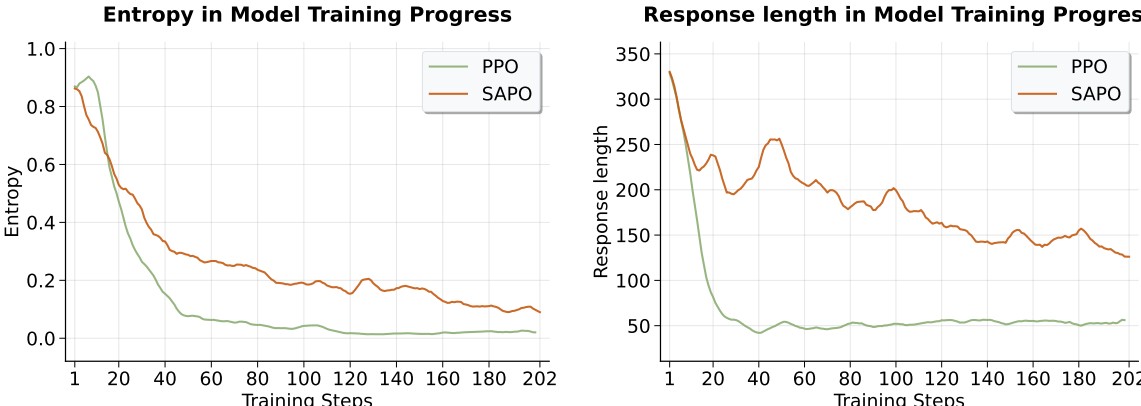

*Figure 7.* Comparison of entropy (left) and response length (right) in training progress between SAPO and PPO.

## C. Further Analysis of Entropy-Based Segmentation

The role of entropy-based segmentation is to provide a statistically grounded proxy for identifying decision-critical transitions. In particular, high-entropy tokens are strongly associated with large value discontinuities:

$$\mathbb{E}[\Delta V_t \mid H_t \geq q] > \mathbb{E}[\Delta V_t], \tag{14}$$

where $\Delta V_t = |V_t - V_{t+1}|$. This indicates that entropy-selected positions are aligned with transitions that are meaningful from the perspective of value prediction.

Even when segmentation is imperfect, it can still be beneficial for value learning. The critic minimizes the squared prediction loss

$$\mathcal{L} = \mathbb{E}\left[\left(V_\theta(h_t) - \hat{R}_t\right)^2\right]. \tag{15}$$

Aggregating tokens within a segment reduces the variance of the return target. If $\hat{R}_m$ denotes the segment-level target and $S_m$ denotes the set of tokens in segment $m$, then under weak dependence among token-level targets,

$$\text{Var}(\hat{R}_m) \approx \frac{1}{|S_m|} \text{Var}(\hat{R}_t). \tag{16}$$

This variance reduction leads to more stable optimization. Missing positions with small $\Delta V_t$ is typically harmless, because such positions contribute little to distinguishing different reasoning branches. In contrast, capturing positions with large $\Delta V_t$ improves credit localization and reduces intra-step variance. Thus, entropy-based segmentation provides a robust approximation that aligns with value-relevant transitions.

The benefit can also be understood from the perspective of policy-gradient variance. In PPO, the token-level policy gradient takes the form

$$g_t = A_t \nabla_\theta \log \pi_\theta(a_t \mid s_t), \qquad g = \mathbb{E}_t[g_t]. \tag{17}$$

At low-entropy steps where the policy is highly confident, i.e., $\pi_\theta(a_t \mid s_t) \approx 1$, the score term $\nabla_\theta \log \pi_\theta(a_t \mid s_t)$ is typically small. Such tokens contribute little to the expected gradient, while still introducing unnecessary token-level noise. Aggregating low-entropy regions into segments therefore performs temporal smoothing through the segment-level advantage

$$\hat{A}_{\text{seg}} = \sum_{i=0}^{\tau-1} \gamma^i r_{t+i} + \gamma^\tau V(s_{t+\tau}) - V(s_t), \tag{18}$$

which suppresses token-level variance and concentrates credit assignment on decision-critical transitions.

# D. Bias Analysis of Geometric-Mean Importance Sampling

We analyze the approximation bias introduced by the geometric-mean importance ratio used in SAPO. This ratio is not an exact importance-sampling correction for the raw segment-level action probability. Instead, it should be understood as a length-normalized correction that trades a bounded approximation bias for lower variance and more stable optimization over variable-length reasoning segments.

Consider a reasoning segment $z_m$ of length $L$. For each token in the segment, define the token-level log-ratio

$$x_t = \log \frac{\pi_\theta(y_t \mid x, y_{<t})}{\pi_{\theta_{\text{old}}}(y_t \mid x, y_{<t})}, \qquad t = 1, \dots, L. \tag{19}$$

The exact segment-level importance ratio is

$$R_m = \frac{\pi_\theta(z_m \mid s_m)}{\pi_{\theta_{\text{old}}}(z_m \mid s_m)} = \prod_{t=1}^{L} \frac{\pi_\theta(y_t \mid x, y_{<t})}{\pi_{\theta_{\text{old}}}(y_t \mid x, y_{<t})} = \exp\left(\sum_{t=1}^{L} x_t\right). \tag{20}$$

SAPO instead uses the geometric-mean ratio

$$\widetilde{R}_m = R_m^{1/L} = \exp\left(\frac{1}{L}\sum_{t=1}^{L} x_t\right). \tag{21}$$

Under the standard PPO small-update regime, assume that $|x_t| \leq \epsilon$ for all tokens. A first-order expansion around $x_t = 0$ gives

$$R_m = 1 + \sum_{t=1}^{L} x_t + O((L\epsilon)^2), \qquad \widetilde{R}_m = 1 + \frac{1}{L}\sum_{t=1}^{L} x_t + O(\epsilon^2). \tag{22}$$

Therefore,

$$R_m - \widetilde{R}_m = \left(1 - \frac{1}{L}\right)\sum_{t=1}^{L} x_t + O((L\epsilon)^2). \tag{23}$$

This shows that the first-order discrepancy between the raw segment ratio and the geometric-mean ratio is controlled by the aggregate token-level policy shift within the segment.

A more explicit bound follows by defining the segment-average log-ratio

$$\mu_m = \frac{1}{L}\sum_{t=1}^{L} x_t. \tag{24}$$

Then

$$R_m = e^{L\mu_m}, \qquad \widetilde{R}_m = e^{\mu_m}. \tag{25}$$

By the mean value theorem,

$$\left|R_m - \widetilde{R}_m\right| = \left|e^{L\mu_m} - e^{\mu_m}\right| \leq e^{L|\mu_m|}(L-1)|\mu_m|. \tag{26}$$

Hence, the approximation error is explicitly bounded by the segment-average log-ratio $\mu_m$ and the segment length $L$. In particular, when the update is small, $\widetilde{R}_m$ remains in an $O(\epsilon)$ neighborhood of 1, whereas the raw segment ratio can drift away from 1 increasingly quickly as $L$ grows.

This establishes that the geometric-mean ratio introduces a bounded perturbation under the small-update regime, rather than an uncontrolled or direction-changing error. The formulation therefore provides a controlled bias–variance tradeoff for variable-length high-level actions: it sacrifices exact segment-level importance sampling in exchange for a length-normalized ratio with substantially improved numerical stability.

# E. Additional Results.

**Results on Qwen3-VL-4B-Thinking.** We evaluate SAPO on Qwen3-VL-4B-Thinking, using the same hyperparameters as the main paper. The result is shown in Tab. 5. SAPO consistently outperforms PPO and GRPO, achieving the best performance, and demonstrates strong generalization to advanced reasoning models.

**Comparison of different segmentation strategies across multiple cross-domain benchmarks.** We further compare our entropy-based segmentation with newline-, step-, and probability-based strategies across multiple cross-domain benchmarks, including scientific reasoning (ScienceQA), cross-domain expert reasoning (MMMU), as well as comprehensive visual-language reasoning (MMBench and MMStar). We consistently observe that, in Tab. 6, our entropy-based segmentation achieves the best overall performance across tasks, indicating better generalization across diverse domains.

*Table 5.* Performance on reasoning benchmarks using Qwen3-VL-4B-Thinking as the base model.

| Method | LogicVista | MathVerse | MathVista | Avg. |
|---|---|---|---|---|
| Base | 53.46 | 63.83 | 75.40 | 64.23 |
| +GRPO | 55.48 | 66.37 | 76.10 | 65.98 |
| +PPO | 56.15 | 63.58 | 76.30 | 65.34 |
| +SAPO | 59.28 | 65.99 | 77.00 | **67.43** |

*Table 6.* Cross-task behavior of different segmentation strategies.

| Strategy | ScienceQA | MMMU | MMBenchV11 | MMStar | Avg. |
|---|---|---|---|---|---|
| Newline-based | 88.10 | 52.77 | 81.66 | 62.80 | 71.33 |
| Step-based | 88.89 | 53.11 | 81.03 | 64.00 | 71.76 |
| Prob-based | 88.49 | 51.78 | 80.93 | 64.00 | 71.30 |
| Entropy-based | 90.08 | 53.22 | 81.61 | 64.53 | **72.36** |

# F. Hyperparameters and Compute Resources.

Our code is based on VeRL (Sheng et al., 2024). We run all experiments on a single node equipped with 8 NVIDIA A100 GPUs (80GB). Detailed hyperparameter configurations are summarized in the Tab. 7.

*Table 7.* Hyperparameters for different training methods on Geo3K.

*(a) SAPO*

| Hyperparameter | Value |
|---|---|
| Total training steps | 200 |
| Train batch size | 512 |
| Mini batch size | 128 |
| Max prompt length | 1024 |
| Max response length | 2048 |
| Lambda | 0.99 |
| Gamma | 1.0 |
| Actor LR | 1e-6 |
| Critic LR | 2e-6 |
| KL penalty in reward | K1 |
| KL coef | 0.001 |
| Eval. temperature | 0.6 |
| Eval. top-$p$ | 0.95 |

*(b) GRPO*

| Hyperparameter | Value |
|---|---|
| Total training steps | 200 |
| Train batch size | 512 |
| Mini batch size | 128 |
| Max prompt length | 1024 |
| Max response length | 2048 |
| Actor LR | 1e-6 |
| KL loss type | K3 |
| KL coef | 0.01 |
| Rollout | 8 |
| Eval. temperature | 0.6 |
| Eval. top-$p$ | 0.95 |

*(c) VC-PPO*

| Hyperparameter | Value |
|---|---|
| Total training steps | 200 |
| Train batch size | 512 |
| Mini batch size | 128 |
| Max prompt length | 1024 |
| Max response length | 2048 |
| Lambda (Actor) | 0.99 |
| Lambda (Critic) | 1.0 |
| Gamma | 1.0 |
| Actor LR | 1e-6 |
| Critic LR | 2e-6 |
| KL penalty in reward | K1 |
| KL coef in reward | 0.001 |
| Eval. temperature | 0.6 |
| Eval. top-$p$ | 0.95 |

*(d) PPO*

| Hyperparameter | Value |
|---|---|
| Total training steps | 200 |
| Train batch size | 512 |
| Mini batch size | 128 |
| Max prompt length | 1024 |
| Max response length | 2048 |
| Lambda | 0.95 |
| Gamma | 1.0 |
| Actor LR | 1e-6 |
| Critic LR | 2e-6 |
| KL penalty in reward | K1 |
| KL coef | 0.001 |
| Eval. temperature | 0.6 |
| Eval. top-$p$ | 0.95 |

