# OpenReview forum: "Segment-Aligned Policy Optimization for Multi-Modal Reasoning"
_ICML.cc/2026/Conference — ICML 2026 regular_

### Official Review · Reviewer_cyXh · 2026-02-14

**Soundness:** 3
**Presentation:** 3
**Significance:** 3
**Originality:** 3
**Overall Recommendation:** 5
**Confidence:** 4

**Summary:**

This paper proposes the SAPO (Segment-Aligned Policy Optimization) method. The core idea is to adjust the granularity of policy optimization in reinforcement learning from the traditional token level or the entire sequence level to a "step" level that is more in line with the human reasoning process. The authors believe that existing methods (such as PPO for token-level optimization and GRPO for sequence-level optimization) have failed to align with the natural structure of reasoning: a complete reasoning step may contain multiple tokens, but these tokens should share the same learning signal. SAPO identifies the boundaries of reasoning steps adaptively based on entropy values, and then performs value estimation, advantage calculation, and importance sampling at the step level. Experiments have achieved more stable and accurate results than the baseline on multimodal and pure text reasoning tasks.

**Compliance With Llm Reviewing Policy:**

Affirmed.

**Final Justification:**

I will keep my initial rates.

**Key Questions For Authors:**

However, there are a few points that make me a little doubtful.

Firstly, the paper states that rewards are only given at the end, and there are no explicit rewards at intermediate steps. Then, would the estimation of the step-level value be difficult to learn due to the sparse rewards? Although the author used GAE for discounted accumulation, the problem of signal attenuation in long reasoning chains may still exist.

Secondly, is the top-k% parameter set to 30% in entropy-based segmentation determined empirically or based on theory? Figure 5 shows that this parameter has a significant impact on the results, but no generalization verification on different datasets has been seen.

In addition, there are no results for the latest model, such as Qwen3.

The final experimental section covered multiple benchmarks, but lacked an analysis of the failure cases: for instance, in which types of questions would SAPO still make mistakes? Is it that some complex reasoning that requires cross-step connections is actually disrupted by the step division? If these details could be supplemented, it would be more persuasive.

**Limitations:**

See Questions. If my question is well solved, I will raise my rating.

**Strengths And Weaknesses:**

The highlight of this paper lies in its precise identification of the problem. Indeed, when observing the PPO training process, it is often noticed that in the same reasoning step, some tokens are rewarded while others are penalized, resulting in a logically awkward situation. The author's design of using entropy values to determine the boundaries of steps is quite ingenious. The experiments also prove that this method is more effective than newline segmentation or fixed-format segmentation. Additionally, the graph of the value function training curve is very intuitive. The value loss of SAPO is indeed much more stable than that of VC-PPO, indicating that step alignment does reduce training noise. What impressed me the most was the comparison chart of response length. PPO quickly reduces the output to only a few tokens, while SAPO can maintain a longer reasoning chain, which should be crucial in long CoT tasks.

See Questions for weakness.

---

> ### Author Rebuttal · Authors · 2026-03-31
>
> Thank you for your thoughtful feedback. We appreciate your recognition of our motivation, entropy-based segmentation, and experimental analyses. Your comments are very encouraging. Below are our responses to your concerns.
> ### **Q1: Analysis of terminal-only rewards.**
> We agree that terminal-only rewards make value estimation challenging due to sparse supervision and long-horizon credit propagation.
>
> Our focus is on a complementary issue: the mismatch between token-level granularity and reasoning structure. Tokens within a reasoning step jointly form subgoals, yet token-level methods assign noisy, inconsistent advantages and different discounting, introducing unnecessary variance and incoherent credit assignment.
>
> SAPO mitigates this issue by aggregating tokens into segments aligned with decision-critical transitions and assigning credit at the segment level. This ensures consistent learning signals within each step, reduces variance, and improves value estimation under sparse rewards.
>
> In addition, our approach is orthogonal to methods with intermediate supervision (e.g., PRM-style rewards), avoiding additional annotation cost while remaining compatible with such methods in future work.
> ### **Q2: Generalization of the top-k parameter**
>
> The top-k=30% setting is chosen empirically rather than derived from a closed-form theory. Our current theoretical intuition is that there exists a nontrivial tradeoff between under-segmentation and over-segmentation: when k is too small, many decision-critical transitions are missed, yielding overly coarse segments and entangled credit signals; when k is too large, many low-value fluctuations or local linguistic noise are also marked as boundaries, causing the step abstraction to degenerate toward noisy token-level updates.
>
> Empirically, the paper already shows this non-monotonic behavior on Geo3K in Fig. 5 in the main paper: top-k values that are too small under-cover decision points, while overly large k values introduce too many low-entropy tokens into segmentation, diluting genuinely informative boundaries. To further verify that this is not specific to Geo3K, we additionally conduct a new sensitivity study on GSM8K. The same interpretation is also consistent with this observation, where performance peaks around k=30 (please see Table 5 below). This supports the view that 30% is not an arbitrary constant, but an empirical operating point within an optimal range balancing missing critical steps and over-fragmenting the trajectory.
>
> **Table 5: Effect of entropy top-k threshold on GSM8K performance.**
> | k     | 10  | 20  | 30  | 40  | 50  | 60  | 70  | 80  | 90  |
> |-------|-----|-----|-----|-----|-----|-----|-----|-----|-----|
> | GSM8K | 56.7| 58.5| 61.4| 59.1| 59.8| 57.8| 57.6| 56.4| 55.5|
>
> ### **Q3: Results for Qwen3 series.**
>
> Thank you for this suggestion. Following your advice, we have extended our experiments to Qwen3-VL-4B-Thinking (Table 1, WMXG Q1) and Qwen3-4B (Table 3, mzEE Q1). All experimental settings are kept consistent with those in the main paper, without additional hyperparameter tuning. Across both settings, SAPO consistently outperforms PPO and GRPO, achieving the best overall performance, indicating improved generalization.
>
> ### **Q4: Analysis of Missing Failure Case Discussion in the Final Experimental Section.**
>
> We agree that adding a failure‑case analysis would make the paper more persuasive. Upon closer inspection, we find that the remaining errors in SAPO are not mainly due to segmentation disrupting cross‑step reasoning. Instead, they can be illustrated with a representative example.
>
> **Early mistakes that lead to a cascade of errors (failing to check intermediate results against given constraints).**
>
> A typical case is the box problem with surface area 54 and volume 23. The model reasons: “Since 23 is a prime number, the possible factorizations are $1×1×23$, so $a=1, b=1, c=23$.” This step already violates the surface‑area constraint $2(ab+bc+ca)=54$, but the model does not verify it and simply continues reasoning from the wrong assumption. In such cases, the issue is not that segmentation breaks the connection between steps; rather, the model makes an incorrect decision at a critical point early on and then follows it consistently.
>
> Overall, this suggests that SAPO’s remaining failures are more likely due to selecting an incorrect intermediate branch or imperfect boundary identification, rather than a general disruption of cross‑step dependencies. In the revision, we will add a more comprehensive taxonomy of failure modes along with qualitative examples.

---

> > ### Author Rebuttal · Reviewer_cyXh · 2026-04-03
> >
> > Thanks!

---

> > > ### Author Response · Authors · 2026-04-03
> > >
> > > We sincerely appreciate your experimental advice and inspiring questions that give us new research directions. We have learned a lot from the review. Thank you again for your effort and supportive feedback.

---

### Official Review · Reviewer_vs2d · 2026-02-23

**Soundness:** 2
**Presentation:** 3
**Significance:** 3
**Originality:** 3
**Overall Recommendation:** 3
**Confidence:** 4

**Summary:**

This manuscript tackles the notoriously difficult credit assignment problem in reinforcement learning for long-horizon reasoning tasks. To bridge the gap between high-variance token-level updates and overly coarse sequence-level rewards, the authors introduce Segment-Aligned Policy Optimization. The core mechanism leverages token prediction entropy, using a top-k% threshold to identify high-uncertainty boundaries and dynamically partition the reasoning trajectory into discrete semantic segments. By elevating the Markov Decision Process from the token level to the segment level, the proposed framework computes values, advantages, and a geometric-mean-based importance sampling ratio over these localized chunks. The empirical evaluation spans both multi-modal benchmarks, such as Geo3K and MathVista, and pure text reasoning tasks, including GSM8K and MATH500, demonstrating consistent gains over standard token-level and sequence-level baselines.

**Compliance With Llm Reviewing Policy:**

Affirmed.

**Final Justification:**

The authors provided a strong rebuttal that addressed most of the technical concerns. However, I am maintaining my Weak Reject score. In its current form, the paper requires a comprehensive structural and aesthetic overhaul that exceeds what a simple revision can fix. I believe the work would benefit from another round of thorough polishing before publication.

**Key Questions For Authors:**

1. Regarding the use of the geometric mean for the importance sampling ratio, can you provide a rigorous theoretical proof demonstrating why this formulation does not introduce a fatal, convergence-breaking bias into the policy gradient? If an approximation bias does exist, what is the strict theoretical error bound for this approximation over long sequences?
2. How exactly does the heuristic handle confident but incorrect reasoning steps? When the model exhibits a low-entropy hallucination during a pivotal mathematical deduction, does your algorithm simply fail to segment it, thereby breaking the credit assignment for that critical error?

3. Please provide a rigorous, end-to-end comparison of training throughput, measured in tokens per second per GPU, and peak memory usage against standard baselines under identical hardware constraints. Exactly what percentage of system overhead is introduced by the real-time entropy sorting and dynamic segment construction?

If you can mathematically formalize the error bounds of your geometric importance sampling and transparently address the system overhead in the rebuttal, I would be open to reconsidering my score.

**Limitations:**

yes

**Strengths And Weaknesses:**

There is no denying that this paper targets a critical and highly relevant bottleneck. Finding a functional middle ground between the noisy token-level updates of standard PPO and the uniformly smoothed credit assignment of group relative policy optimization is arguably one of the most valuable pursuits in current RLVR research. Furthermore, the empirical validation is commendable, extending beyond standard text-based math reasoning to encompass multi-modal architectures like Qwen2.5-VL, which effectively demonstrates the method's cross-domain applicability.

However, when evaluated against the rigorous algorithmic standards expected at a top-tier venue, the proposed framework relies far too heavily on brittle heuristics and exhibits glaring theoretical gaps in its reinforcement learning formulation, leading to my Weak Reject recommendation.

The most critical flaw lies in the dynamic segmentation mechanism itself. Relying purely on a global top-k% entropy threshold to define logical boundaries is an inherently noisy and conceptually flawed assumption. High token entropy only indicates vocabulary-level predictive uncertainty; it does not equate to a complete, semantically coherent reasoning step. Consider a scenario where the model suffers from a "confident hallucination," producing a mathematically incorrect intermediate result with extremely low entropy. In this case, the segmentation mechanism entirely misses this critical decision boundary, causing the subsequent penalty signal to be improperly diluted. Furthermore, hardcoding a global hyperparameter to handle prompts of vastly different lengths, complexities, and domain distributions is empirically fragile.

On the theoretical front, the mathematical formulation of the segment-level importance sampling ratio is highly problematic. To prevent the variance explosion typical of multiplicative importance weights over long segments, the authors resort to taking the geometric mean of the token-level ratios in their formulation. While this brutal engineering trick intuitively stabilizes the loss, it fundamentally destroys the unbiased nature of the policy gradient estimate. The manuscript provides absolutely no formal mathematical proof or theoretical error bound to guarantee that this geometric approximation converges within a standard actor-critic framework. Without this rigor, the core objective function reads less like a principled algorithmic innovation and more like a forced hack to make the math cooperate.

Finally, the authors conspicuously evade the reality of system-level computational overhead. Continuously computing full-sequence entropy, performing global sorting, and dynamically constructing computation graphs for segment boundaries during the rollout phase inevitably throttles the training throughput. For researchers focused on the systems and efficiency side of large language model training, omitting a transparent discussion on the extra memory footprint and wall-clock time overhead feels intellectually dishonest.

---

> ### Author Rebuttal · Authors · 2026-03-31
>
> ### **Q1: Theoretical implications and bias analysis of geometric-mean importance sampling.**
>
> We agree our formulation introduces a bounded, controlled bias relative to the exact segment-level IS ratio, not a convergence-breaking one. Under the PPO-style clipped surrogate, its effect is further limited. Our goal is to use a length-normalized segment ratio to reduce variance and stabilize optimization, since the raw ratio can scale too aggressively with segment length, causing premature clipping and unstable updates, as noted in our response to **WMXG Q3.**
>
> Formally, let the token-level log-ratio within segment $m$ be
> $$x_t=\log \frac{\pi_\theta(y_t\mid x,y_{<t})}{\pi_{\text{old}}(y_t\mid x,y_{<t})},$$
> Then the exact unbiased segment-level ratio is
> $$
> R_m = \frac{\pi_\theta(z_m\mid s_m)}{\pi_{\text{old}}(z_m\mid s_m)}
> = \prod_{t=1}^{L} \frac{\pi_\theta(y_t\mid x,y_{<t})}{\pi_{\text{old}}(y_t\mid x,y_{<t})}
> = \exp\left(\sum_{t=1}^{L}x_t\right).
> $$
> while SAPO uses the geometric-mean ratio
> $$
> \tilde R_m = R_m^{1/L} = \exp\left(\frac{1}{L}\sum_{t=1}^{L}x_t\right).
> $$
>
> Under the PPO small-update regime, where |x_t|\le \epsilon, a first-order expansion around x_t=0 gives
> $$R_m = 1+\sum_{t=1}^{L}x_t+O((L\epsilon)^2),\qquad \tilde R_m = 1+\frac{1}{L}\sum_{t=1}^{L}x_t+O(\epsilon^2),$$
> and therefore
> $$R_m-\tilde R_m = \left(1-\frac{1}{L}\right)\sum_{t=1}^{L}x_t+O((L\epsilon)^2).$$
> This shows that the first-order discrepancy between the raw ratio and the geometric-mean ratio scales with segment length.
>
> A more rigorous bound can be obtained by defining
> $$\mu_m=\frac{1}{L}\sum_{t=1}^{L}x_t,$$
> so that
> $$R_m=e^{L\mu_m},\qquad \tilde R_m=e^{\mu_m}.$$
> Then
> $$|R_m-\tilde R_m| = |e^{L\mu_m}-e^{\mu_m}| \le e^{L|\mu_m|}(L-1)|\mu_m|.$$
> Hence, the approximation error is explicitly controlled by the segment-average log-ratio $\mu_m$. In particular, the raw segment ratio drifts away from 1 increasingly fast as segment length grows, while the geometric-mean ratio is length-normalized and remains in an $O(\epsilon)$ neighborhood of 1.
>
> Thus, under the standard small-update regime, the resulting gradient differs from the raw surrogate only by a bounded perturbation, rather than a direction-changing error. We therefore view our formulation as a controlled bias–variance tradeoff for variable-length high-level actions, rather than a strict IS identity.
>
> ### **Q2: Entropy-based segmentation to low-entropy erroneous reasoning steps**
> Although confident-but-wrong steps can occur, they are not the dominant pattern for reasoning boundaries. In multi-step reasoning, pivotal transitions more often align with competing continuations and high uncertainty, making entropy a useful proxy. Our analysis shows that high-entropy tokens are enriched for large value discontinuities: the $Lift$ score increases with entropy quantile $q$ and exceeds 1 in the high-entropy tail (**Fig3**), indicating that entropy preferentially captures decision-critical transitions.
>
> We do not claim entropy is a perfect semantic oracle. Low-entropy but incorrect steps may not be isolated, since entropy reflects local uncertainty rather than correctness. SAPO is not designed to detect every error token; instead, it forms step-wise optimization units that are more coherent than token-level updates and more localized than sequence-level optimization. Thus, entropy-based segmentation is a statistically effective interface for credit assignment, consistent with our motivation that token-level signals are noisy within a step while sequence-level signals are too coarse.
>
> By grouping tokens into segment-level units, SAPO reduces inconsistent or conflicting signals within a step, which is supported by the value learning dynamics in Fig.6.
>
> Therefore, SAPO does not aim to perfectly recover every semantic boundary, but to provide **a better bias–variance tradeoff** for reasoning credit assignment. The enrichment of high-entropy tokens for decision points, together with improved value-training stability.
>
> ### **Q3: Comparison of training throughput.**
> Relative to PPO, SAPO adds only lightweight overhead: (i) token-level entropy (if not already available), (ii) a top-k selection, (iii) boundary mask construction, and (iv) a single pass to aggregate token-level rewards/values into segment-level quantities for GAE/returns, with shared boundaries for policy and value updates. This involves simple sequence-level tensor operations, without extra rollout or model inference.
>
> At the same time, SAPO reduces downstream credit-assignment cost. PPO performs GAE over all $T$ tokens, while SAPO recurses over $M$ segments with $M \ll T$. Thus, although SAPO adds a one-time entropy/top-k/mask step, it shortens the advantage/return recursion, making the net overhead smaller than segmentation alone suggests.
>
> Empirically, under identical hardware, PPO uses 66.3 GB vs. 66.4 GB for SAPO, with throughput 508 vs. 504 tokens/s/GPU—only ~0.15% extra memory and 0.8% slowdown.

---

> > ### Author Rebuttal · Reviewer_vs2d · 2026-04-01
> >
> > I appreciate the detailed rebuttal, which successfully addresses all of my concerns. However, given the substantial amount of rewriting required to properly integrate these updates into the main text, I feel the manuscript is not yet ready for publication. Therefore, I will maintain my original score.

---

> > > ### Author Response · Authors · 2026-04-03
> > >
> > > We deeply appreciate the time and effort dedicated to your detailed and  informative review. We are glad that we were able to address all the concerns.
> > > We would like to clarify that the required updates are localized and do not involve substantial rewriting of the main paper.
> > > - The theoretical clarification on importance sampling can be incorporated as a short paragraph in the method section, with full derivations provided in the appendix.
> > > - We note that the discussion of confident-but-wrong steps is a lightweight conceptual addition, which can be seamlessly integrated into Sec. 4.2 alongside Fig. 3, further reinforcing the rationale of the approach.
> > > - The system overhead analysis (throughput and memory) naturally fits into the experimental section as a small table; additional details can be included in the appendix and referenced in the main text accordingly, highlighting that SAPO introduces only lightweight overhead compared to PPO.
> > > All in all, these changes are incremental and can be cleanly integrated without altering the core structure or contributions of the paper. More detailed analyses will be placed in the appendix to preserve the clarity and focus of the main text.
> > >
> > > Given that all technical concerns have been addressed and the revisions are limited in scope, we sincerely hope these clarifications will contribute to a positive assessment of our work. We would also be happy to address any further questions or concerns.

---

### Official Review · Reviewer_mzEE · 2026-03-10

**Soundness:** 3
**Presentation:** 3
**Significance:** 2
**Originality:** 3
**Overall Recommendation:** 4
**Confidence:** 5

**Summary:**

This paper proposes Segment-Aligned Policy Optimization (SAPO), whose main idea is to treat coherent reasoning segments, rather than individual tokens or full sequences, as the basic units of reinforcement learning optimization. The method formulates reasoning as a step-wise MDP and performs value estimation, advantage computation, and importance sampling at the segment level, leading to policy updates that are better aligned with the semantic structure of reasoning steps. The paper also introduces an entropy-based adaptive segmentation mechanism that uses high-entropy tokens to dynamically identify likely reasoning boundaries without requiring manual annotations or predefined output formats. Experimental results on multimodal reasoning and text-only mathematical reasoning benchmarks show that SAPO generally outperforms PPO, GRPO, VC-PPO, and some prior segment-level methods, while also yielding more stable training and more consistent value learning.

**Compliance With Llm Reviewing Policy:**

Affirmed.

**Key Questions For Authors:**

NA

**Limitations:**

yes

**Strengths And Weaknesses:**

Strengths
1. The entropy-based adaptive segmentation mechanism is practically appealing, as it does not rely on manual annotations or rigid output formatting.
2. The reported results show fairly consistent performance gains across benchmarks, together with improved training stability and value learning behavior.

Weaknesses
1. Although the paper evaluates both multimodal and text-only reasoning, the text-only evaluation remains somewhat limited, mainly focusing on GSM8K and MATH500. Including more recent and challenging benchmarks that are now commonly used, such as AIME2025, would help more comprehensively assess the effectiveness and generalization of SAPO in text-only complex reasoning settings.
2. While the paper proposes an entropy-based adaptive segmentation mechanism and shows improvements over several baseline segmentation strategies, the analysis of this component could be further strengthened. For example, a more systematic comparison with newline-based, step-based, and probability-based segmentation in terms of segmentation quality, boundary stability, and cross-task behavior would help clarify the source of its advantages and its applicability.

---

> ### Author Rebuttal · Authors · 2026-03-31
>
> Thank you for your thoughtful feedback. We appreciate your recognition of our entropy-based adaptive segmentation as a practical, annotation-free solution, as well as the consistent performance gains and improved training stability of SAPO across benchmarks. We also value your suggestions on further experiments and clarity, and respond to them below.
>
>
>
> ### **Q1: Effectiveness and Generalization of SAPO in Text Reasoning**
>
>
>
> We train **Qwen3-4B** on a mixture of GSM8K and MATH, and evaluate it on the out-of-distribution benchmark **AIME2025**. Notably, due to time constraints, we reuse the same hyperparameters as in the main paper without task-specific tuning.
>
>
>
> As shown in **Table 3 below**, SAPO achieves the best performance on **AIME2025,** outperforming both GRPO and PPO. Since AIME2025 is an out-of-distribution benchmark relative to the GSM8K and MATH training data, the improvement is naturally more modest than in in-domain settings. Nevertheless, the result still demonstrates that SAPO can provide effective optimization and retain reasonable generalization ability in text-only complex reasoning. All experimental hyperparameters are kept consistent with those in the main paper, and the evaluation metric is Avg@32.
>
>
> **Table 3: Performance of Qwen3-4B on AIME25**
>
> | Method   | AIME2025 |
> | -------- | -------- |
> | Qwen3-4B | 47.4     |
> | GRPO     | 55.0     |
> | PPO      | 54.5     |
> | SAPO     | 56.9     |
>
>
>
> ### **Q2: Analysis of the entropy-based adaptive** **segmentation** **mechanism.**
>
>
>
> Firstly, we clarify that **segmentation** **quality** in our setting is defined by how well boundaries align with **decision-critical transitions**, i.e., positions with large value changes (ΔV). Entropy-based segmentation achieves higher quality under this criterion, as high-entropy tokens are statistically enriched for large ΔV (**Fig. 3 in the main paper**), indicating better alignment with return-relevant structure. In contrast, newline- and step-based methods rely on surface-level or explicit formatting, which do not necessarily correspond to decision boundaries, while probability-based methods often capture local uncertainty (e.g., rare tokens or formatting noise) that is not correlated with decision impact. As a result, entropy-based segmentation provides more effective boundaries for credit assignment.
>
>
>
> In terms of **boundary stability**, entropy-based segmentation is more robust to variations in generation style and task format, as it depends on the model’s uncertainty over competing continuations rather than surface structure. In contrast, newline- and step-based methods are sensitive to formatting differences (e.g., whether reasoning is explicitly structured), leading to inconsistent boundaries across samples or tasks. Probability-based methods are also less stable, as low-probability tokens may arise from local noise or rare token usage, resulting in fragmented or spurious boundaries. Empirically, we observe that entropy-based segmentation maintains consistent performance under a fixed top-k across diverse benchmarks **(please see Table 2 in WMXG Q4)**, suggesting more stable boundary behavior.
>
>
>
> In terms of **cross-task behavior**, we further compare entropy-based segmentation with newline-, step-, and probability-based strategies across multiple cross-domain benchmarks under the same setting used in Table 4 in the main paper, including scientific reasoning (ScienceQA), cross-domain expert reasoning (MMMU), as well as comprehensive visual-language reasoning (MMBench and MMStar). We consistently observe that, **in Table 4 below**, entropy-based segmentation achieves the best overall performance across tasks. Combining with Table 3 in the main paper, we can conclude that entropy-based segmentation not only performs well on in-domain tasks, but also maintains strong performance on out-of-domain benchmarks, indicating better generalization across diverse domains.
>
>
>
> **Table 4: Cross-task behavior of segmentation strategies**
>
> | Strategy      | ScienceQA | MMMU | MMBenchV11 | MMStar | Avg.  |
> |---------------|-----------|------|------------|--------|-------|
> | Newline-based | 88.10     | 52.77| 81.66      | 62.80  | 71.33 |
> | Step-based    | 88.89     | 53.11| 81.03      | 64.00  | 71.76 |
> | Prob-based    | 88.49     | 51.78| 80.93      | 64.00  | 71.30 |
> | Entropy-based | 90.08     | 53.22| 81.61      | 64.53  | 72.36 |

---

> > ### Author Rebuttal · Reviewer_mzEE · 2026-04-05
> >
> > Thank you for the detailed response, I will maintain my original score.

---

> > > ### Author Response · Authors · 2026-04-07
> > >
> > > Thank you again for your time and consideration. We are encouraged to see that your concerns have been fully addressed, and we are grateful for your feedback throughout the review process.

---

### Official Review · Reviewer_WMXG · 2026-03-12

**Soundness:** 2
**Presentation:** 3
**Significance:** 2
**Originality:** 2
**Overall Recommendation:** 4
**Confidence:** 5

**Summary:**

This paper aims to explore the optimisation of large language models in multi-step reasoning tasks through reinforcement learning, focusing on improving the credit assignment method during training. The core concept proposed in the study is SAPO, which updates policies based on reasoning segments rather than individual tokens or entire sequences, in order to enhance training stability and reasoning performance.

**Compliance With Llm Reviewing Policy:**

Affirmed.

**Final Justification:**

The authors only provide descriptive explanations for Q2 and Q3, without adequate theoretical justification. The reported improvements seem to stem from empirical parameter tuning rather than rigorous theoretical support.


After I submitted my acknowledgment, the authors provided an additional round of rebuttal, in which they included a detailed proof addressing the points I raised. In principle, such content should have been presented in the first-round rebuttal response, rather than being submitted after my acknowledgment. This also does not justify the attack on me by Reviewer cyXh. Theoretically, I reserve the right to form my own judgment. However, considering the authors’ second-round rebuttal in its entirety, I believe my concerns have been resolved, and I am willing to revise my score accordingly.

Good luck

**Key Questions For Authors:**

1. The proposed SAPO strategy should be validated on stronger reasoning models, such as Qwen3-VL-2B/4B/8B thinking models. Since Qwen2.5-VL may be relatively weaker at this time point, improvements could be easier to obtain. Qwen2.5-VL exhibits notably weak reasoning capabilities, especially in its 3B and 7B variants, as reasoning performance was not a primary focus during its development phase.
2. If the segment division is incorrect, all subsequent step-wise values, advantages, and IS will be based on incorrect boundaries. However, the author's segmentation mechanism essentially still takes the top-k% high-entropy tokens as boundaries. Why is this an effective semantic step detector? Through ablation experiments, performance is best around k=30, but continuing to increase or decrease it results in a significant drop, indicating that the method is sensitive to the segmentation hyperparameter. High-entropy tokens do not necessarily equal reasoning step boundaries; they may simply indicate local expression uncertainty, format uncertainty, or even pure decoding noise. The paper only demonstrates that a certain entropy heuristic happens to be effective, which significantly undermines the general applicability of the method.
3. The theoretical analysis of SAPO needs to be supplemented. Why does step-wise MDP lead to better credit assignment than token-level MDP under a terminal-only reward setting? Why should the segment-level importance ratio use the geometric mean, does this preserve reasonable estimation properties, and does it introduce additional bias? The paper describes entropy-based segmentation as a principled interface, but the term principled here lacks theoretical basis and appears more like an empirical heuristic.
4. The author only trained using Geo3K train and then tested on mathematical benchmarks, but the topk in the paper is extremely sensitive to the k parameter, and the effectiveness of the method in more general reasoning tasks is not certain.

If the author can fully demonstrate the effectiveness of SAPO and provide sufficient theoretical analysis, I would be very willing to increase my score.

**Limitations:**

yes, the authors adequately discuss the limitations and potential negative societal impact of their work.

**Strengths And Weaknesses:**

Strengths: The paper has a clear structure, proposes a new post-training strategy, and designs certain experiments for verification.

Weaknesses: The paper lacks sufficient validation of the method's effectiveness on a wide range of reasoning tasks and more powerful reasoning models, and also lacks accurate theoretical analysis of the method. For details, please refer to Key Questions For Authors.

---

> ### Author Rebuttal · Authors · 2026-03-31
>
> Thanks for your constructive feedback. We appreciate your recognition of our paper’s clarity, the novelty of our post-training strategy, and our experimental efforts. Below are our responses to your concerns.
>
> ### **Q1: Performance on stronger reasoning models.**
>
> We evaluate SAPO on **Qwen3-VL-4B-Thinking (Table 1)**, using the same hyperparameters as the main paper. SAPO consistently outperforms PPO and GRPO, achieving the best performance. Together with the text-only results on **Qwen3-4B (Table 3, mzEE Q1)**, this demonstrates strong generalization to advanced reasoning models.
>
> **Table 1: Performance of Qwen3-VL-4B-Thinking.**
>
> | Method  |LogicVista|MathVerse|MathVista|  Avg.  |
> |---------|----------|---------|---------|--------|
> | Baseline|53.46     |63.83    |75.40    | 64.23  |
> | +GRPO   |55.48     |66.37    |76.10    | 65.98  |
> | +PPO    |56.15     |63.58    |76.30    | 65.34  |
> | +SAPO   |59.28     |65.99    |77.00    | 67.43  |
>
> ### **Q2: Validity and Robustness of Entropy-Based** **Segmentation**
>
> We analyzed the enrichment ratio $Lift$ of Δvalue on high-entropy tokens w.r.t. the entropy quantile threshold $q$ (**Fig. 3**). As training progresses, the region where $Lift >1$ expands toward lower $q$, indicating that large Δvalues increasingly concentrate on high-entropy tokens. Since Δvalue reflects changes in expected future reward, these tokens correspond to decision-critical points, making them suitable segmentation boundaries for aligned credit assignment and value estimation.
>
> In addition, SAPO can be viewed as an adaptive credit assignment mechanism centered on Δvalue. Tokens within a segment correspond to a shared decision branch; aggregating them into a unified optimization unit avoids inconsistent or conflicting credit signals within a step. Thus, SAPO reduces intra-step variance while preserving sensitivity to key decisions, achieving **a better bias–variance tradeoff**.
>
> What's more, this Δvalue-aligned segmentation improves value estimation stability. Token-level values are often noisy due to local fluctuations, while SAPO anchors updates at decision transitions, focusing learning on positions that affect future rewards, leading to more stable value estimates (**Fig. 6**).
>
> Finally, we validate the robustness of the top-k hyperparameter across datasets (**Table 5 in cyXh Q2**). There exists an optimal range of top-k values that reflects **a nontrivial tradeoff between under- and over-segmentation**: too small k misses key decisions, while too large k introduces noisy boundaries and degrades into token-level updates.
>
> ### **Q3: Theoretical analysis of step-wise** **MDP** **and segment-level importance ratio.**
>
> We acknowledge that the key challenge under terminal-only rewards is accurate credit propagation. In reasoning, tokens are not independent decisions but jointly form subgoals, and token-level assignment introduces unnecessary variance, with tokens in the same step receiving noisy or even conflicting advantages. Step-wise MDP addresses this by grouping tokens into segments aligned with decision-critical transitions and assigning credit at this level, reducing within-step variance while preserving sufficient resolution to distinguish reasoning paths, leading to more stable and effective credit assignment.
>
> While the exact unbiased segment-level importance ratio equals the product of token-level ratios, it is poorly conditioned in long-horizon settings due to multiplicative scaling with segment length, leading to high variance. Our geometric-mean formulation introduces a bias–stability tradeoff by normalizing in log-probability space, producing more stable updates across segments of varying lengths. We further apply PPO clipping to control bias. A theoretical error bound is provided (**vs2d Q1**). Empirically, the unbiased variant performs worse (45.79 < 46.90), exhibits more volatile entropy dynamics, and converges to higher value loss (0.2 > 0.05), indicating that stable optimization is more effective in practice.
>
> ### **Q4: More general reasoning tasks and parameter sensitivity analysis.**
>
> We transfer the same top-k (30%) setting without retuning to broader benchmarks: scientific reasoning (ScienceQA), commonsense multimodal reasoning (MMMU), and comprehensive visual-language reasoning (MMBench, MMStar). As shown in **Table 2**, SAPO achieves the best overall performance, indicating that its effectiveness does not rely on dataset-specific tuning of k, but generalizes well across different domains and task types. We also conduct extra top-k parameter **sensitivity ablation on GSM8K (cyXh Q2)**.
>
> **Table 2: Performance on general reasoning tasks**
> | Method      |ScienceQA|MMMU |MMBenchV11|MMStar|Avg. |
> |-------------|---------|-----|----------|------|-----|
> | Qwen2.5VL-7B|88.34    |52.67|81.23     |62.73 |71.24|
> | GRPO        |88.34    |51.67|81.35     |63.67 |71.26|
> | PPO         |88.79    |50.44|81.09     |64.00 |71.08|
> | SAPO        |90.08    |53.22|81.61     |64.53 |72.36|

---

> > ### Author Rebuttal · Reviewer_WMXG · 2026-04-03
> >
> > The authors only provide descriptive explanations for Q2 and Q3, without adequate theoretical justification. The reported improvements seem to stem from empirical parameter tuning rather than rigorous theoretical support. Therefore, I will maintain my original score.

---

> > > ### Author Response · Authors · 2026-04-04
> > >
> > > Thanks for your response. Below are our responses.
> > > # Q2:
> > > Entropy-based segmentation does not need to perfectly recover all semantic steps. It serves as a statistically grounded proxy for identifying decision-critical transitions. High-entropy tokens are strongly associated with large value discontinuities:
> > > $$
> > > \mathbb{E}[\Delta V_t \mid H_t \ge q] > \mathbb{E}[\Delta V_t],
> > > $$
> > > indicating alignment with meaningful reasoning transitions.
> > > Even when segmentation is imperfect, it remains beneficial from a value-learning perspective. The critic minimizes the loss
> > > $$
> > > \mathcal{L} = \mathbb{E}\big[(V_\theta(h_t) - \hat{R}_t)^2\big],
> > > $$
> > > and segment-level aggregation effectively reduces target variance:
> > > $$
> > > \mathrm{Var}(\hat{R}_m) \approx \frac{1}{|S_m|}\\mathrm{Var}(\hat{R}_t),
> > > $$
> > > leading to more stable optimization (as shown in Fig. 6).
> > > Notably, missing positions with small $\Delta V_t$ is typically harmless, since they contribute little to distinguishing different reasoning branches. In contrast, capturing large $\Delta V_t$ improves credit localization and reduces intra-step variance.
> > > Thus, entropy-based segmentation provides a robust approximation aligned with value-relevant transitions and remains empirically effective.
> > >
> > > From an optimization viewpoint, the benefit can also be understood through gradient variance reduction. In PPO, the policy gradient is
> > > $g_t = A_t \\nabla_\theta \log \pi_\theta(a_t \mid s_t)$, $g = \mathbb{E}_t [g_t]$
> > >
> > > At low-entropy steps where $\pi(a_t \mid s_t) \approx 1$, we have $\nabla_{\theta} \log \pi_{\theta} \approx 0$. This means such tokens contribute little to the expected gradient, while still introducing unnecessary variance.
> > >
> > > Aggregating low-entropy regions into segments performs temporal smoothing via the segment-level advantage
> > > $\hat A_{\mathrm{seg}} = \sum_{i=0}^{\tau-1} \gamma^i r_{t+i} + \gamma^\tau V(s_{t+\tau}) - V(s_t)$,
> > > which suppresses token-level noise and concentrates credit assignment on decision-critical transitions.
> > >
> > > For top-k selection, the optimal range (e.g., around 30%) reflects a natural bias–variance tradeoff:
> > > small $k$ leads to under-segmentation and biased credit propagation, while large $k$ increases variance toward token-level PPO.
> > > Thus, $k$ controls the effective granularity of the MDP abstraction rather than acting as an ad-hoc heuristic.
> > > Empirically, a fixed $k$ generalizes across domains without retuning (Table 2 in WMXG). A sensitivity analysis on GSM8K (Table 5 in cyXh) with $k\in[10,90]$ shows stable performance over a wide range $k\in[20,50]$, with peak performance around $k=30$, confirming that SAPO operates within a robust interval.
> > >
> > > # Q3:
> > > First, SAPO can be interpreted as an adaptive temporal abstraction, in the spirit of Semi-MDPs [1], where variable-length reasoning chunks replace fixed token-level transitions. From this perspective, we do not arbitrarily “break” the token-level MDP; instead, we lift it to a higher-level Semi-MDP whose temporal units are adaptive reasoning segments.
> > >
> > > This design is especially beneficial under terminal-only rewards:
> > > token-level PPO must propagate rewards across all $T$ tokens, whereas SAPO only propagates credit through $M$ segments, where typically $M\ll T$.
> > > SAPO therefore shortens the effective horizon and performs value estimation over semantically more coherent partial states, rather than over incomplete token prefixes. This provides a principled explanation for why step-wise value and advantage estimation is better conditioned than token-level estimation in long-CoT settings.
> > >
> > > Second, this step-wise abstraction yields a clear variance-reduction effect. In token-level PPO, the gradient estimator takes the form
> > > $$
> > > \sum_t \hat A_t \nabla_\theta \log \pi_\theta(y_t\mid x,y_{<t}).
> > > $$
> > > In long reasoning traces, many low-entropy tokens correspond to locally deterministic continuations. These tokens provide little useful decision signal, yet still inject token-level noise into advantage propagation.
> > > By grouping locally coherent tokens into segments and assigning a shared segment-level credit signal, SAPO reduces intra-step variance and focuses optimization on decision-relevant transitions.
> > > The advantage thus comes not from entropy perfectly capturing semantics, but from defining an adaptive temporal abstraction with a superior bias–variance tradeoff compared to fixed token-level updates.
> > >
> > > Third, regarding the segment-level importance-sampling (IS): we acknowledge that our geometric-mean formulation is not a strictly unbiased IS for the raw segment action. We proved its **theoretical bound in vs2d Q1**. [Here's also the full proof.](https://share.jotbird.com/quiet-serene-chili-pepper)
> > > Therefore, we view the geometric-mean ratio as a controlled bias–variance tradeoff for variable-length reasoning segments.
> > >
> > > # Reference
> > > [1] Richard S. Sutton, Doina Precup, and Satinder Singh. 1999. Between MDPs and semi-MDPs: a framework for temporal abstraction in reinforcement learning. *Artif. Intell.* 112, 1–2 (Aug. 1999), 181–211.

---

### Decision · Program_Chairs · 2026-04-30

**Decision:**

Accept (regular)

**Comment:**

The paper proposes SAPO, which formulates RL post-training as a segment-level MDP, uses high-entropy tokens as segment boundaries, and estimates values at the segment level during training. Experimental results on multimodal and mathematical reasoning benchmarks show that SAPO outperforms PPO and GRPO. All reviewers agree on the significance and originality of the method. The authors have provided thorough responses to concerns about theoretical analysis and robustness/generalizability of the experimental results. Given the reviews and rebuttal, we recommend acceptance and encourage the authors to incorporate these clarifications and additional results into the final version.